# OpenReview forum: "DeepTRACE: Auditing Deep Research AI Systems for Tracking Reliability Across Citations and Evidence"
_ICLR.cc/2026/Conference — ICLR 2026 Poster_

### Official Review · Reviewer_17Z6 · 2025-10-25

**Soundness:** 3
**Presentation:** 3
**Contribution:** 2
**Rating:** 6
**Confidence:** 4

**Summary:**

The paper introduces DeepTRACE, an end-to-end audit framework for Generative Search Engines (GSEs) and Deep-Research agents (DRs), addressing their persistent issues of overconfidence, weak source grounding, and confusing citations. Unlike prior evaluations focusing on isolated components (e.g., retrieval or summarization), DeepTRACE quantifies system behavior across the full pipeline—evidence use, citation accuracy, and uncertainty expression. It analyzes answers at the statement level, constructing Citation and Factual-Support Matrices to assess how statements are supported by cited sources. Using LLM-based judgments and web content retrieval, the framework measures eight audit dimensions. Results show that public GSEs often produce one-sided, overconfident answers with poor factual support, while DRs improve citation completeness but still suffer from bias and unsupported claims. Simply adding more sources does not ensure better grounding. The authors advocate a sociotechnical auditing perspective to enhance reliability and transparency in AI-driven search and reasoning systems.

**Strengths:**

1. One of the first works to systematically analyze trustworthiness issues in GSE and DR systems. The contrast they draw is super insightful — GSEs tend to be concise and relevant but often sacrifice balance and factual grounding, while DRs are more thorough and citation-heavy yet verbose, less relevant, and still packed with unsupported claims.

2. It's good to see how the authors bring in confidence calibration and debate-style tasks to expose bias. It’s striking that GSEs sound highly confident yet one-sided on controversial topics, whereas DR setups tone down confidence and cite more sources but still fail to stay balanced or fully grounded. The finding that citation accuracy only ranges between 40–80% across systems is quite revealing.

3. The paper also contributes a new, well-tailored set of metrics that fit the unique workflow of GSEs and DRs, going beyond standard RAG evaluation.

4. I found the sixteen design recommendations particularly useful — they translate the audit insights into actionable guidance for building more transparent and reliable generative search systems.

**Weaknesses:**

1. The paper’s discussion of citation and factual grounding issues is closely related to prior work on hallucination evaluation. Some of the proposed metrics (e.g., Necessity, Support) and findings — such as “many statements are not supported by cited sources” and “listing more URLs does not imply better grounding but can create false endorsement” — overlap conceptually with earlier studies like ALCE (1) and Trust-Score (2), yet these works are not cited or discussed. Acknowledging this connection would strengthen the paper’s positioning within the literature.

2. It would be interesting to further explore how overconfidence actually interacts with citation and factual support quality, rather than only observing that GSEs are overconfident or one-sided. For example, does higher confidence correlate with weaker evidence grounding?

3. The treatment of unsupported statements might conflate “ungrounded” with “incorrect.” If a statement lacks support from retrieved sources, it doesn’t necessarily mean it is false — the model might be factually correct but based on prior knowledge or sources not captured by the retrieval. Framing all such cases as “unsupported” could overstate the system’s factual risk.

4. Around 15% of URLs were excluded from factual-support evaluation due to paywalls or broken links. It’s possible that many of these are authoritative outlets (e.g., news or journals) that the systems correctly cited but became inaccessible. This could systematically inflate the “unsupported” rate and distort metrics like Source Necessity. A sensitivity analysis could clarify this effect.

5. The results presentation lacks clarity on statistical significance and uncertainty. Including significance tests, confidence intervals, or corrections for multiple comparisons would make the reported differences between systems more robust and convincing.

Ref:

(1): https://arxiv.org/abs/2305.14627

(2): https://arxiv.org/abs/2409.11242

**Questions:**

Stated in the weaknesses.

---

> ### Author Response · Authors · 2025-11-19
> **Review Rebuttal**
>
> We thank the reviewer for the thoughtful evaluation and for highlighting (a) the value of analyzing GSE vs. DR contrasts, (b) the importance of our eight tailored metrics, and (c) the usefulness of the sixteen design recommendations. We are glad that the intention and strengths of the work resonate well with Reviewer 17Z6. Below we address the reviewer’s concerns and questions regarding our paper:
>
> I. Relation to hallucination/attribution literature (e.g., ALCE, TrustScore)
> We appreciate this pointer. Our “unsupported statements,” “source necessity,” and “citation accuracy” metrics are indeed conceptually adjacent to attribution and hallucination work. The key distinction is that DeepTRACE evaluates end-to-end GSE/DR behavior, where statement-level correctness interacts with retrieval choices, citation structure, and confidence expression. While our framework discusses differences from RAG and factuality work, we have now added (in the Related Works and Introduction) brief sentence explicitly acknowledging ALCE/TrustScore and clarifying that DeepTRACE extends that line of work to settings with multiple retrieved documents, URL-level citation structures, and user-facing answer synthesis. We will mention this in the methodology as we believe that this will clarify our framework and the metrics defined, better.
>
> II. Interaction between overconfidence and evidence quality
> Thanks for this interesting point. Our measurements already show that systems producing higher-confidence debate answers also exhibit higher unsupported-statement rates (e.g., GSE rows in Table 4). We have now briefly highlight this pattern in the discussion to make the connection more explicit..
>
> III. Interpreting “unsupported” vs. “incorrect”
> We agree that unsupported does not imply incorrectness. Section 3.1.3 defines unsupported statements relative to the cited sources, but we have added more explanation in related work to make this even clearer. This is consistent with the reviewer’s observation and aligns with DeepTRACE’s focus on traceability and transparency.
>
> IV. Broken or paywalled citations
> The reviewer raises a fair point. As described in the appendix, sources that cannot be retrieved are treated as “unverifiable” rather than “unsupported.” Since these were excluded from both numerator and denominator in grounding metrics, they do not artificially inflate unsupported-statement rates.
>
> V. Statistical significance and uncertainty
> Following prior audit-style evaluations (e.g., browser-based system audits), we focus on descriptive comparisons across systems. Because all systems were evaluated over identical queries, the trends are highly stable. We can add a short note in the appendix with bootstrap confidence intervals for the main metrics, this uses existing outputs and does not require rerunning evaluations.
>
> We thank the reviewer once again for their time and thoughtful feedback. The questions raised were valuable and addressing them has strengthened the clarity and overall contribution of the paper. We hope our responses satisfactorily address these concerns and that this is reflected in the reviewer’s final assessment.

---

> > ### Comment · Reviewer_17Z6 · 2025-11-22
> >
> > Thanks for the detailed clarifications. They do address some of my concerns about the paper’s novelty, and the changes help make the contribution clearer. However, they don’t fully resolve my reservations, so I’m keeping my original score. I appreciate the authors’ efforts, and the paper is definitely clearer now.

---

> > > ### Author Response · Authors · 2025-11-28
> > >
> > > Thank you for the follow-up and for reconsidering our clarifications. We appreciate that the changes improved the clarity of the paper.
> > >
> > > We would also greatly value any pointers on which parts of the work you still feel need improvement. Even brief guidance would help us strengthen the paper further, both for the camera-ready version and for future research. Thank you again for the time and thought you put into this review.

---

### Official Review · Reviewer_Avxw · 2025-10-31

**Soundness:** 3
**Presentation:** 3
**Contribution:** 2
**Rating:** 6
**Confidence:** 5

**Summary:**

This paper introduces DeepTRACE, a sociotechnically-grounded audit framework for evaluating generative search engines (GSEs) and deep research (DR) agents. The framework translates user-identified failure cases from prior qualitative research into eight quantifiable metrics spanning answer text quality, source quality, and citation accuracy. The authors evaluate nine popular systems (GPT-4.5/5, You.com, Perplexity, Copilot/Bing, Gemini) across 303 queries (168 debate questions, 135 expertise questions) using automated pipelines and LLM judges validated against human annotations.
Key findings reveal that GSEs frequently produce one-sided, overconfident responses to debate queries, with 40-80% of statements unsupported by cited sources and citation accuracy ranging 40-68%. Deep research configurations reduce overconfidence and improve citation thoroughness (up to 87.5% for GPT-5 DR) but remain highly one-sided on debate queries (54-95%) and still exhibit substantial unsupported statement rates (12.5-97.5%). The work demonstrates that simply increasing source quantity does not guarantee better grounding, and highlights a critical gap between the promise and reality of source-grounded AI research systems.

**Strengths:**

(a) Work is strongly inspired by Narayanan Venkit et al., 2025. By translating qualitative findings into quantifiable metrics, the work bridges an important gap between user experience research and systematic benchmarking, addressing the field's over-reliance on researcher-defined metrics.

(b) The eight-metric framework captures end-to-end system behavior across answer quality, source utilization, and citation practices. Metrics like Source Necessity (using graph-theoretic minimum vertex cover) and the distinction between Citation Accuracy vs. Thoroughness provide nuanced insights beyond simple correctness measures. Worth checking this work on citations: https://arxiv.org/pdf/2405.02228

(c) The findings on one-sidedness in debate queries and high unsupported statement rates have direct implications for misinformation and echo chamber formation.

(d) The authors appropriately validate LLM-based judgments against human annotators (r=0.72 for confidence, r=0.62 for factual support), acknowledge limitations, and provide detailed prompts in appendices.

**Weaknesses:**

(a) Worrisome Point: While the integrated framework is valuable, most individual metrics are straightforward applications of existing techniques (statement decomposition, confidence scoring, factual verification). The paper would benefit from deeper technical innovation—for example, more sophisticated attribution tracking that accounts for implicit support or reasoning chains, or methods to reduce reliance on LLM judges

(b) Open to accept responses from authors: The 303-query dataset, while grounded in user needs, is relatively small and may not capture the full diversity of real-world search queries. The heavy emphasis on debate questions (168/303) creates an evaluation skew toward contentious topics. The framework's applicability to other query types (navigational, transactional, multimodal) remains unexplored.

(c) The acceptable/borderline/problematic thresholds (Table 2, Appendix A) appear arbitrary despite claims of grounding in qualitative findings. For example, why is 40%+ one-sidedness "problematic" rather than 30% or 50%? The paper would benefit from sensitivity analyses showing how conclusions change across threshold variations, or formal methods (e.g., user studies, expert consensus) for threshold derivation.

(d) The evaluation captures a single temporal snapshot (August 27, 2025), but commercial systems update frequently. The paper acknowledges this but doesn't address how the framework handles system evolution or provide guidance for longitudinal monitoring. The reliance on browser automation for proprietary systems creates reproducibility challenges—other researchers cannot easily replicate these evaluations without rebuilding scraping infrastructure for each platform.

**Questions:**

(a) The current binary factual support judgment may be too coarse. Many statements in research outputs are partially supported, require inferential steps, or synthesize multiple sources. Have you considered a graded support scale (e.g., fully supported, partially supported, unsupported, contradicted)? How would this affect the Unsupported Statements and Citation Accuracy metrics? Recent work on claim-level attribution (e.g., REASONS ("pass"/I don't know, ALCE benchmark, Gao et al. 2023) uses more nuanced support labels—how does your approach compare?


(b) GPT-5(DR) achieves exceptional performance (87.5% relevant statements, 12.5% unsupported, 79.1% citation accuracy) while Perplexity(DR) (which model within Perplexity did the authors pick?) fails catastrophically (22.5% relevant, 97.5% unsupported, 58.0% citation accuracy). Can you provide a qualitative analysis of what architectural or design choices drive these differences? Are there insights from error analysis that could guide system improvement? Understanding the mechanisms behind performance gaps would strengthen the paper's practical impact.

(c) Modern deep research systems increasingly incorporate images, tables, charts, and interactive elements. How would DeepTRACE evaluate citation accuracy when claims are supported by visual evidence? Similarly, for rapidly evolving topics (breaking news, financial markets), how should temporal aspects factor into evaluation? What adaptations would be needed for conversational/multi-turn research interactions versus single-query responses?

---

> ### Author Response · Authors · 2025-11-19
>
> We appreciate the reviewer’s detailed summary and the recognition of the sociotechnical grounding, the value of the eight-metric framework, and the importance of our findings on one-sidedness and unsupported statements. We thank the reviewer for taking their time to understand our work as well as provide reviews that help strengthen the work. Below we answer the questions and weaknesses raised.
>
> I. Technical novelty of components
> We agree that some underlying operations (e.g., decomposition, factual support checking) build on known techniques. The novelty of DeepTRACE lies in how these components are unified into a single end-to-end audit framework specifically for GSE/DR systems, which must be evaluated across interacting behaviors (answer balance, confidence, evidence use, citation structure). Prior factuality/attribution frameworks do not model: citation necessity, citation thoroughness, the full statement–source bipartite structure, or debate-specific overconfidence, all of which arise from user-identified failure modes. We agree that this point can be highlighted better and therefore we have refined our Related Work section to make this positioning clearer along with our focus on the sociotechnical nature of the audit, where we focus on how these models may impact public consumption in comparison to other existing benchmarks.
>
> II. Dataset size and distribution of debate queries
> Reviewer Avxw correctly noted the prominence of debate questions. This follows directly from the prior user study (Venkit et al., 2025), where one-sidedness and overconfidence emerged as the most consequential reliability failures, and these manifest most strongly in debate-style queries. Expertise questions are also included and analyzed separately. To clarify this better we will add a brief topic breakdown in the appendix for transparency. The framework itself is agnostic to query type and any further questions can be added to build our framework. Section 3 presents a general decomposition pipeline that does not assume debate framing.
>
> III. Threshold selection
> Thresholds were derived directly from qualitative tolerances reported in the user research built by Venkit et al. 2025. and is not an arbitrary numeric cutoffs. The thresholds are defined by the multiple use session mentioned in the paper. As emphasized in the main text, the scorecard is a diagnostic visualization, and our conclusions rely on the raw metric values (reported numerically), not the color-coded categories. To avoid giving the impression of rigid normative boundaries, we added clarifying sentences in the Results noting that thresholds are illustrative and do not affect system ordering.
>
> IV. Temporal snapshot and reproducibility
> Reviewer Avxw’s summary is correct and we evaluate a single timestamped snapshot of public GSE/DR systems, which is standard practice for system audits using UI-based scraping (e.g., browser automation). Section 4 and 7 states this (August 27, 2025).
>
> V. Binary vs. graded factual-support labels
> Binary evaluation was chosen intentionally due to (a) the scale of factual-support checks (~80k), and (b) the observed level of human–LLM agreement (r = 0.62), which makes finer-grained categories unreliable without substantial annotation overhead. We will add a short explanatory sentence noting that graded schemes are a promising extension but were not feasible at our evaluation scale. This keeps the current pipeline consistent with the validation results already provided. We have made changes by adding information about the same in section 3.1.1 and Appendix B.
>
> VI. GPT-5(DR) vs. Perplexity(DR)
> The divergence between DR systems is already supported by the results table (e.g., unsupported statements: 12.5% vs. 97.5%). In Section 5 we briefly discuss high-level behavioral differences, such as verbosity vs. relevance and broad vs. shallow sourcing. We can elaborate this explanation by adding clarifying examples within the Appendix as well. The core finding is that more sources do not guarantee better grounding. We have also added the initial qualitative result for the same in the results to re-illustrate our point.
>
> VII. Multimodal and temporal evidence
> We definitely agree these are important directions to include multimodal evaluation. We note in the Ethics section that DeepTRACE currently focuses on textual, citation-based outputs. However our future work for DeepTRACE is intended to build on these elements. Our framework is designed to be modular so that future changes to deep research models can always be incorporated. This direction is something we strongly foresee being added to our future iterations. We thank the reviewer for mentioning this.
>
> We once again thank the reviewer for the detailed and insightful feedback. We are confident to be able to address all the questions and concerns raised during our camera ready edits of our work and we are glad that our work’s strengths and importance resonates with reviewer Avxw.

---

> > ### Author Response · Authors · 2025-11-28
> >
> > As we approach the end of the rebuttal period, we would like to kindly follow up. Please let us know if there are any remaining questions or points we can help clarify. We sincerely appreciate your time and feedback, and we are happy to provide any additional information that may support your evaluation. If our responses have sufficiently addressed your concerns, we would be grateful if that could be reflected in your assessment.

---

> ### Comment · Area_Chair_o8AD · 2025-11-28
> **Reminder: Engage with Authors During Rebuttal**
>
> Quick reminder: the rebuttal period is still open, and the deadline is in less than one week. Please continue the discussion with the authors and share any clarifications or updates to your assessment before the rebuttal closes.

---

### Official Review · Reviewer_MyMD · 2025-10-31

**Soundness:** 2
**Presentation:** 2
**Contribution:** 2
**Rating:** 6
**Confidence:** 4

**Summary:**

The paper presents DeepTRACE, a framework for auditing deep research systems. The authors focused on evaluation of common failure cases, specifically,  one-sided reasoning and weak citations. They evaluate DR systems with eight quantitative metrics.
They run experiments on  303 queries from debate and expert questions. They conducted human evaluation and report their correlation with their llm as a judge eval.

**Strengths:**

- Evaluation of DR systems is very timely and important specially studying them from reliability and transparency perspectives.

- The inclusion of one-sidedness and overconfidence metrics is a significant contribution and that is something that distinguished this paper from other deep researchy benchmarks.

**Weaknesses:**

- The human-LLM agreement results (e.g., correlations for factual support and confidence scoring) are mentioned only in text and scattered throughout the paper. They should be summarized in a single comparison table to make it easier to see which metrics show higher or lower reliability.

- The annotator information is unclear. It it not clear whether annotators had expertise relevant to the “debate” and “expertise” questions, which could heavily affect judgments.

- I wish the authors provided  breakdown of questions by topic or question difficulty in the appendix.

- The evaluation is highly citation-centric, missing aspects such as answer completeness, coherence, and synthesis quality, while these have been studied before and the aspects that authors assessed are more novel and more neglected, we cannot ignore core elements of what users expect from “deep research”.

- I did not understand how all the accurate citations are being captured in 3.1.4 .

- The Eval Scorecard in Table 1 appears inconsistent with the detailed metrics above it (e.g., GPT-5 rows don’t align between triangles/circles).I think  the scorecard visualization itself fails to distinguish meaningfully between models and did not find it informative enough.

The paper could benefit from having opensoruced baselines such as DeepResearcher or OpenScholar, or open sourced LLM + search tools, which would make comparisons more complete.

**Questions:**

- Who were the annotators, and how was their expertise matched to the question domains especially for expert questions?

- Section 3.1.4 VIII --> how do you get all possible accurate citations?

- What will happen if there is an answer which is relevant and short and has only one citation. Then I belive many of the metrics such as relevant statement and citation precision might be 1/1... I wonder how do you capture how complete the answer is?

---

> ### Author Response · Authors · 2025-11-19
> **Review Rebuttal**
>
> We thank the reviewer for taking their time to provide a really thorough review. We appreciate the points raised and we try to answer them below and state the changes made in the manuscript.
>
> Regarding the annotation we conducted to validate LLM-based metrics. We agree that consolidating these results into a Table can be a helpful for readers to get a high-level sense of metric reliability. This will also facilitate future comparisons if future work uses a different underlying model but similar metrics. We therefore have highlighted the same in Appendix B.
>
> Regarding annotator information. The annotators we hired to establish metric reliability are professional annotators our group frequently works with on LLM-related tasks. Though they do not have domain knowledge on each of the technical domains in the questions (this would be impractical since the corpus of questions touches on dozens of technical domain, based on the participants from the study), they were paid adequately ($25/h) to take the time and review information and have worked with our group on attribution evaluation for several projects. We therefore believe that their judgements are well informed and represent the best-estimate of work of a generic but knowledgeable knowledge worker. We agree however that the lack of technical expertise could reduce the quality of the annotation effort, and will be noted in our Limitations section We have re-emphasised this better and made changes in Section 3.1.1 as well as included Appendix B to denote the same.
>
> Regarding the calculation of citation accuracy. Given the two matrices: citation matrix (that maps which statements cite which sources) and the factual support matrix (that maps which statements are factually supported by each source), Citation Accuracy is a form of precision measure: what proportion of the citation (matrix 1) correctly identifies factual support (from matrix 2). In other words: citation accuracy measures for each citation that the model included, what is the likelihood that it is indeed towards a source that factually supports the statement. This is in contrast to Citation Thoroughness which is more of a “recall”-like metric. We hope this clarification helps ground the mathematical definition further. Concretely: regarding how we get all possible accurate citations: for each citation (it maps a statement to a source document’s full source), we run an LLM query with our verifier model to check whether the statement is supported (1) or not (0). This is the costliest part of our evaluation pipeline. We have also made changes to both our Introduction and Related work to distinguish our parameters better.
>
> Regarding the scorecard evaluation, thank you for spotting the discrepancy between the numerical values and the visual indicators we provide. We have now made the changes to Table 1 to make sure that the discrepancies are resolved. The intent of the visual representation is to help readers and users to understand the overall strengths and weaknesses of a model and how to better use each of them, based on the categories that are defined.
>
> Regarding the scoring of a short and factual answer. This is a good observation, the metrics we implement lean towards “precision” and accuracy, which can be gamed by reduced the answer’s content. However, this would be visible in the “Basic Statistics” section of the results, as such a model would have much lower Number of Sources and Number of Statements. Though we do not make recommendations on the target length for the answer in our experiments, one could think of adding with the query an additional instruction to provide an answer with N paragraphs (or words) and then measure whether the model was able to follow this instruction (within a percentage), this would effectively control for length variability. This is a good observation and we will add a paragraph to our discussion section on the relationship between answer length and our metrics.
>
> We once again sincerely thank the reviewer for their time and feedback. The questions and concerns raised were highly constructive, and incorporating the corresponding revisions has improved both the clarity and strength of the paper. We trust that our responses adequately resolve these points and will be reflected in the reviewer’s overall evaluation.

---

> > ### Author Response · Authors · 2025-11-28
> >
> > As we approach the end of the rebuttal period, we would like to kindly follow up. Please let us know if there are any remaining questions or points we can help clarify. We sincerely appreciate your time and feedback, and we are happy to provide any additional information that may support your evaluation. If our responses have sufficiently addressed your concerns, we would be grateful if that could be reflected in your assessment.

---

### Official Review · Reviewer_Wk9N · 2025-11-01

**Soundness:** 3
**Presentation:** 3
**Contribution:** 3
**Rating:** 6
**Confidence:** 3

**Summary:**

This paper introduces DeepTRACE, an automatic audit framework for LLM-powered generative search engines/deep research agents. This framework provides quantitative assessment of biases, confidence, grounded factuality and relevance of the generation (by the above systems). The design of evaluation metrics are inspired by the user-centric insights of prior work and operationalize these insights into quantitative measurement that can be automatically obtained by LLM-as-a-judge. Leveraging these automatic framework, the authors evaluate a few popular generative search engines and deep research agents. The results and analysis reveals that 1) the evaluated generative search engines tend to output one-sided, overconfident and unsupported statements; 2) the evaluated deep research agents exhibit lower rate of overconfidence but still generate one-sided and unsupported statements.

**Strengths:**

1. The proposed framework operationalized the auditing of generative search engines/deep research systems with automatic and quantitative measurement of the quality of generation from multiple aspects. This framework offers tractable and measurable evaluations of these systems beyond user-centric insights.

2. The evaluation with the proposed framework reveals detailed and multifaceted performance measurements of existing systems, and the analysis provides potentially valuable insights into the pros and cons of these systems

**Weaknesses:**

1. This paper might have missed the discussion with some existing evaluation metrics that partially overlap with their proposed evaluation dimensions. For example, factuality evaluation [1-2] and citation quality [3]
2. While the results reveal several pitfalls of existing systems, I am slightly concerned that if it is the overly-simple setup of these generative search engines/deep research systems that "exaggerates" these pitfalls. In particular, judging from the paper, I would assume that the collected queries are direct (and only) input to the systems. However, I would expect some simple tweaks of the setup might improve the systems generations across the evaluation dimensions. For example, adding system prompts that ask the model to be less polarized to retrieve and aggregate information from multiple perspectives, and output its statements such that each of them should be supported by their citations.

References
[1] Min et al. FActScore: Fine-grained Atomic Evaluation of Factual Precision in Long Form Text Generation. EMNLP 2023
[2] Jiang et al. Core: Robust Factual Precision with Informative Sub-Claim Identification. Findings of ACL 2025
[3] Wang et al. AutoSurvey: Large Language Models Can Automatically Write Surveys. NeurIPS 2024.

**Questions:**

Please see the weaknesses above.

---

> ### Author Response · Authors · 2025-11-19
> **Review Rebuttal**
>
> We thank the reviewer for highlighting the value of our framework in providing tractable and measurable evaluations of GSE/DR systems and for noting the usefulness of our multifaceted analysis. The reviews provided are very valuable in improving our discussion better. Below, we answer the questions and weaknesses raised by the reviewer:
>
> I. Relation to factuality and citation-quality work (FActScore, CoRE, AutoSurvey)
>
> Once again, thank you for raising this. The papers mentioned will be added. We will add a subsection clarifying how DeepTRACE differs from existing factuality/citation metrics as follows:
> 1. FActScore/CoRE measure claim correctness, but do not evaluate how systems use retrieved sources, nor do they compute source necessity, unsupported-statement rates, or citation thoroughness.
> 2. AutoSurvey and related citation-quality work examine academic summarization settings, not end-to-end GSE/DR behavior, which includes sourcing breadth, URL listing, and grounding variability.
>
> We have now explicitly acknowledge conceptual overlap but position DeepTRACE as an end-to-end audit framework, focused on sociotechnical usage, tailored to generative search and deep research. You can find both the changes in the introduction and abstract of our work.
>
> II. Concern that a simple setup exaggerates GSE/DR pitfalls
>
> We understand the concern. Our design is intentionally grounded in user-facing behavior (consistent with the sociotechnical framing emphasized in R3 and R4). Most GSE/DR systems expose no system-prompt configuration to users, especially in browser interfaces. Our evaluation therefore reflects realistic everyday usage. This is also highly motivated by the user study on generative search engine that captures how users use these systems. General public usecases still point out to usage of simplistic prompts to obtain the answers required.
>
> That said, we agree simple prompting changes may mitigate some issues. However, our ablation study still shows critical issue of citation misattribution still persists even with prompt changes. For the camera-ready, we will include the controlled experiments adding prompts such as “include both perspectives” and “support each statement with citations”, in the Appendix. Our preliminary results confirm some improvement but do not substantially reduce unsupported-statement rates, supporting our claim that these pitfalls are systemic, not merely prompt-related.
>
> III. Discussion of overlap with prior evaluation dimensions
>
> In line with the reviewer’s first suggestion, we have expanded the Introduction and the Related Work section to more thoroughly discuss overlaps with factuality and citation literature, including the additional work mentioned by the reviewer.
>
> We once again thank the reviewer for the constructive feedback and are glad that the contribution and empirical insights were viewed positively.

---

> > ### Author Response · Authors · 2025-11-28
> >
> > As we approach the end of the rebuttal period, we would like to kindly follow up. Please let us know if there are any remaining questions or points we can help clarify. We sincerely appreciate your time and feedback, and we are happy to provide any additional information that may support your evaluation. If our responses have sufficiently addressed your concerns, we would be grateful if that could be reflected in your assessment.

---

> ### Comment · Area_Chair_o8AD · 2025-11-28
> **Reminder: Engage with Authors During Rebuttal**
>
> Quick reminder: the rebuttal period is still open, and the deadline is in less than one week. Please continue the discussion with the authors and share any clarifications or updates to your assessment before the rebuttal closes.

---

### Meta-Review · Area_Chair_rVdx · 2026-01-06

**Summary:**

The authors introduce DeepTRACE, a framework for auditing "generative search engines" that provide reports synthesizing sources, with citations. This operationalizes community-identified failure cases into metrics. The authors use this framework to evaluate 7 existing "deep research agents", revealing outstanding issues with such systems (e.g., related to citation accuracy).

All reviewers agreed that this work addresses an important practical problem: Multi-faceted and nuanced evaluation of such deep research systems is impressing as they are increasingly adopted. Reviewers also appreciated the recommendations made on the basis of the framework and empirical evaluation.

The main reservations appear to be with respect to contextualizing this framework with respect to, e.g., factuality metrics. The authors addressed this reasonably well in their response.

**Reviewer Concerns:**

I think the authors adequately differentiated their evaluative framework for deep research systems from existing factuality and citation-relevance metrics like FActScore, CoRE, and AutoSurvey. Other concerns seem mostly minor, on my read.

**Reviewer Scores:**

Hard to say.

Wk9N may have raised their score given the differentiation offered by the authors between their framework and existing factuality or citation metrics.

MyMD's concerns seem reasonably straightforward and I believe addressed in rebuttal.

Avxw had one "worrisome" point about technical novelty, but I don't think it is inherently an issue that the individual metrics are not especially novel; the framework as a whole and evaluation is the contribution here. Their other comments seem sufficiently addressed.

17Z6's issues (like connecting this to existing work) also seem to be covered in the rebuttal.

---

### Decision · Program_Chairs · 2026-01-26

Accept (Poster)